# Influence of the Flow Rate in an Automated Microfluidic Electronic Tongue Tested for Sucralose Differentiation

**DOI:** 10.3390/s20216194

**Published:** 2020-10-30

**Authors:** Maria L. Braunger, Igor Fier, Flávio M. Shimizu, Anerise de Barros, Varlei Rodrigues, Antonio Riul

**Affiliations:** 1Department of Applied Physics, “Gleb Wataghin” Institute of Physics (IFGW), University of Campinas (UNICAMP), Campinas SP 13083-859, Brazil; malubraunger@yahoo.com.br (M.L.B.); flamakoto@yahoo.com.br (F.M.S.); varlei@ifi.unicamp.br (V.R.); 2Quantum Design Latin America, Campinas SP 13080-655, Brazil; igor@qd-latam.com; 3Laboratory of Functional Materials, Institute of Chemistry (IQ), University of Campinas (UNICAMP), Campinas SP 13083-970, Brazil; anerisedebarros@gmail.com

**Keywords:** microfluidic electronic tongue, impedance spectroscopy, sucralose, flow rate

## Abstract

Incorporating electronic tongues into microfluidic devices brings benefits as dealing with small amounts of sample/discharge. Nonetheless, such measurements may be time-consuming in some applications once they require several operational steps. Here, we designed four collinear electrodes on a single printed circuit board, further comprised inside a straight microchannel, culminating in a robust e-tongue device for faster data acquisition. An analog multiplexing circuit automated the signal’s routing from each of the four sensing units to an impedance analyzer. Both instruments and a syringe pump are controlled by dedicated software. The automated e-tongue was tested with four Brazilian brands of liquid sucralose-based sweeteners under 20 different flow rates, aiming to systematically evaluate the influence of the flow rate in the discrimination among sweet tastes sold as the same food product. All four brands were successfully distinguished using principal component analysis of the raw data, and despite the nearly identical sucralose-based taste in all samples, all brands’ significant distinction is attributed to small differences in the ingredients and manufacturing processes to deliver the final food product. The increasing flow rate improves the analyte’s discrimination, as the silhouette coefficient reaches a plateau at ~3 mL/h. We used an equivalent circuit model to evaluate the raw data, finding a decrease in the double-layer capacitance proportional to improvements in the samples’ discrimination. In other words, the flow rate increase mitigates the formation of the double-layer, resulting in faster stabilization and better repeatability in the sensor response.

## 1. Introduction

An electronic tongue (e-tongue) is a multisensory analytical device mimicking the gustatory system to analyze complex liquids, based on the physical or chemical transduction of signals [1]. With the aid of computational and statistical tools, raw data is transformed into patterns that create a “fingerprint” of each analyte [2,3]. Different types of e-tongues have been reported in numerous applications [4,5,6,7,8], being more recently integrated into microfluidic devices [9,10] for synergistic advantages such as the increased frequency of testing, drastic reduction of sample and discharge volumes, decreasing waste, and overall cost of the analysis. Moreover, the possibility of controlling the sample flow rate results in faster stabilization and better repeatability of the sensor response. However, measurements on this type of device may be time-consuming due to several operational steps with tubing and cabling of each sensing unit to the electrical testing equipment for data acquisition. Researchers usually choose one flow rate to develop the sensing analysis based on empirical tests [11,12,13]. Therefore, to the best of our knowledge, no systematic experiments were previously conducted to evaluate distinct flow rates in an e-tongue system in the literature.

One of the most challenging operations in e-tongues is the non-trivial task of distinguishing substances having a sweet taste due to the non-electrolyte nature of molecules responsible for sweetness [14]. That is an essential assignment in food quality and masking unpleasant tastes (generally bitter) from active compounds commonly used in pharmaceuticals. In the past 15 years, several studies were performed with e-tongues to discriminate sweeteners and evaluate their suppression effects [14,15,16,17,18]. Sucralose is an artificial sweetener and one of the most commonly used sugar substituents nowadays [19,20]. The literature reports e-tongues engineered to evaluate taste-masking and suppression effects on sucralose and other sweeteners [15,16,17]. Briefly, Dyminski et al. tested 11 formulations of commercial sweeteners using an e-tongue based on conducting polymer films deposited onto gold interdigitated electrodes [15]. The device responded successfully to seven distinct formulas (sucrose, acesulfame K, aspartame, cyclamate, sucralose, acesulfame K with aspartame, and sucralose with cyclamate), with responses reasonably correlated with a human panel (trained human judges), except for formulations containing higher ionic concentration that could not be perceived by the panelists. Legin et al. used an e-tongue based on PVC membranes to evaluate the bitterness suppression of a pharmacological ingredient by adding sucralose, aspartame with acesulfame K, and grape juice [16]. The latter was the most effective sweetener for taste masking, followed by the mixture of aspartame and acesulfame K, while sucralose was the least effective. Choi et al. developed an e-tongue to evaluate the taste-masking effect of neohesperidin dihydrochalcone, sucrose, sucralose, and aspartame on four different pharmaceutical drugs [17]. The authors concluded that the taste-masking impact was dependent on the type of active pharmaceutical ingredients and that the e-tongue can detect small concentration changes of a chemical in solution.

Commercial sweeteners usually contain several ingredients, such as thickening agents, acidity regulators, flavoring, and preservatives. However, commercial sweeteners with the same sweetening compound are supposed to deliver pretty much the same taste, at least for most consumers. In this context, instead of evaluating the distinction of different sweeteners, we performed some tests to check the e-tongue ability to distinguish sucralose-based sweeteners, as the local market sells them as the same food product. Here, we have no intention of discovering the source of the differences among commercial samples but evaluating whether it is possible to distinguish them using a microfluidic e-tongue device.

Beyond the difficulty of distinguishing sucralose-based sweeteners once they present pretty much the same taste, we employ emerging and inexpensive techniques to develop a robust e-tongue system for rapid and efficient data acquisition. Our goal is to simplify multiple acquisitions with the least possible external interferences in the e-tongue measuring system within the context discussed above. We were able to perform tests on commercial sweeteners with an automated device faster than usual methods. Therefore, a systematic study of the influence of the analyte flow rate in the microfluidic e-tongue was feasible, thus confirming the hypothesis that controlling the flow rate reduces the impact of double-layer effects, resulting in faster stabilization and better repeatability of the sensor response.

## 2. Materials and Methods

### 2.1. Collinear Electrodes and Automated Multiplexing

The core of the sensing setup is based on four collinear, gold-plated interdigitated electrodes (IDEs) embedded on a printed circuit board (PCB), fabricated by TEC-CI Circuitos Impressos (São Paulo, SP, Brazil) according to the authors instruction. All sensing units of the e-tongue device lie in a single straight microchannel (detailed ahead), aiming for rapid data acquisition. The IDEs were designed with four pairs of digits having 5 mm length, 0.2 mm width, and 0.2 mm spacing. In a separate electronic assembly, a digitally controlled analog multiplexing system was designed and built to independently route the signal of each IDE to a pair of coaxial connectors. This unit features an inexpensive Arduino^®^ Nano microcontroller module, communicating via USB to a PC unit. The Arduino^®^ powers an array of small-signal relays through a Darlington driver integrated circuit (ULN2003) from Texas Instruments (Dallas, TX, USA). The relays are arranged in a cascade fashion, preventing signal superimposition. The inherent “break-before-make” behavior of the relays is also advantageous to avoid cross-talking of the routed signals. Besides rapid data acquisition, the use of this e-tongue setup mitigates errors originating from manual operations, wiring of individual electrodes, and wear and tear of electrical contacts.

### 2.2. Layer-by-Layer Deposition onto Collinear Electrodes

Our e-tongue device requires an array of at least four sensing units with distinct chemical compositions and electrical responses to analyze samples [3]. The choice of the sensing units is based on the idea of mimicking the biological system; that is, grouping electrical signals from distinct sources that can then be easily classified into specific patterns. The best way to do it is to choose materials having different electrical responses to enable a fingerprint of the solutions analyzed, which can be processed with statistical and computational techniques. Three electrodes were modified with nanostructured materials, and one was left uncovered (bare IDE), thus producing four distinct sensing units [21]. We chose the layer-by-layer (LbL) technique for the film assembly as it is a versatile and straightforward way to form multilayered nanostructures on solid substrates, relying on physical interactions of oppositely charged polyelectrolytes [22]. Here, copper phthalocyanine-3,4′,4″,4′″-tetrasulfonic acid tetrasodium salt (CuTsPc), montmorillonite clay (MMt-K), poly(3,4-ethylenedioxythiophene)-poly(styrenesulfonate) (PEDOT:PSS), and poly(diallyldimethylammonium chloride) solution (PDDA) were purchased from Sigma-Aldrich and used as received. These materials were dispersed in deionized water obtained from an Arium^®^ comfort Sartorius purifier and the LbL films were fabricated by sequentially alternating their aqueous solutions onto the IDEs to form 10 bilayers of PDDA/CuTsPc, PDDA/MMt-K, and PDDA/PEDOT:PSS, similar to that reported in [11,23].

The electrodes’ collinear arrangement prevented the straightforward fabrication of LbL film on the PCB by regular dipping or fluidic methods. Accordingly, we developed a hollow structure to enable individual LbL deposition onto each IDE, as shown in Figure 1. A casting mold was 3D-printed onto an acrylic slide by the fused deposition modeling (FDM) method, as shown in Figure 1a. After that, polydimethylsiloxane (PDMS) elastomer was prepared using the base and curing agents (10:1) of a Sylcap™ 284-F kit from MicroLubrol (Clifton, NJ, USA). The viscous PDMS liquid was kept for 1 h under vacuum to remove air bubbles and then poured into the 3D-printed mold (Figure 1b). Once the PDMS structure was cured, it was removed from the mold and reversibly sealed by mechanical pressure on the PCB using an acrylic plate and stainless-steel screws, allowing the LbL deposition. Finally, the collinear e-tongue comprises one bare IDE and three others covered with PDDA/CuTsPc, PDDA/MMt-K, and PDDA/PEDOT:PSS LbL films.

### 2.3. Fabrication of Elastomer Channels and Device Assembling

The research presented here had no intention of studying delicate microfluidic devices; therefore, we avoid previously reported photolithography or 3D-printing stereolithography methods for microchannel fabrication [10,11]. Endorsing the cost reduction philosophy, we 3D-printed a microchannel mold onto an acrylic slide by FDM, as shown in Figure 2a. The microchannel mold was sketched using Autodesk Inventor software with 500 µm width, 500 µm height, and 4 cm length. PDMS was prepared as mentioned in Section 2.2, being cast into the mold for the microchannel fabrication. After curing the PDMS microchannel, the inlet and outlet ports were made using a biopsy punch. Finally, the PDMS microchannel was reversibly sealed by mechanical pressure using an acrylic plate and stainless steel screws on the PCB having the four collinear sensing units (details in Figure 2). Although simple, this procedure ensures a leak-tight operation and further evaluation or characterizations of the sensing units.

### 2.4. Impedance Measurements on the Automated Microfluidic e-Tongue

The e-tongue device was tested in four distinct brands of liquid sucralose-based sweeteners (Zero-Cal^®^, Linea^®^, Finn^®^, and Adocyl^®^) commercially available in the local market. In addition to the nearly identical taste in all samples, the challenge here is the differentiation of a flavor that is already difficult to distinguish in most systems found in the literature once sweet taste substances show non-electrolyte properties [14]. Aqueous solutions of each brand were prepared at 10 μL/mL, the equivalent of ~5 drops in one shot of espresso coffee. Impedance data were acquired while the solutions were flowing through the microchannel using a custom-made syringe pump from 0 to 15 mL/h flow rates for comparison purposes. Additionally, measurements were performed in a particular flow rate (5 mL/h) for sucralose dispersed in deionized water and in espresso coffee in five different concentrations. The coffee analysis was made with capsules of illy^®^ iperEspresso classic roast and deionized water. The flow rate changes, impedance measurement, and electrode multiplexing were controlled by computer software developed by the authors to assist in the systematic evaluation of their influence on the sensing response (Figure 3).

Impedance measurements were performed at 25 mV amplitude in the frequency range 1–10^6^ Hz using a Solartron 1260A impedance/gain-phase analyzer. The e-tongue device coupled to the multiplexer is shown in Figure 4. The sensing units were automatically routed to the testing equipment during the experiment using computer software developed by the authors (Figure 3). As all sensing units are within the same microchannel, three independent measurements of each analyte were automatically performed, and the operator only manipulated the e-tongue setup to change analytes and clean the channel. The device was rinsed with 10 mL of deionized water between each analyte sample to ensure no cross-contamination.

Figure 5 shows a schematic representation of the e-tongue setup used to evaluate the sucralose-based sweeteners in different flow rates. Briefly, the procedure automatically controlled by the software is: (i) the syringe pump establishes the flow rate into the microfluidic device; (ii) the multiplexer selects the sensing units on the PCB; (iii) the impedance analyzer performs the measurements at each sensing unit three times for statistical validation; (iv) the flow rate changes according to the sequence program, and the procedure is repeated for the same analyte. Summarizing what relies on manual operations: (i) cleaning between different analytes with deionized water, and (ii) loading the syringe filled with another analyte in the pump. The same procedure was used to evaluate the sweeteners at distinct water and coffee concentrations, but this time at one specific flow rate. It is worth mentioning that the response of a control solution (distilled water) is always evaluated after measurements in each analyte sample to assess drift or cross-contamination effects [24], as performed in [11].

The raw data (electrical impedance measurements) were dimensionally reduced by principal component analysis (PCA) available in Orange Data Mining software (version 3.26) [25], projecting multidimensional data into a 2D Euclidean space for better data interpretation and visualization. The quality of data discrimination was evaluated through the silhouette method [26], in which the silhouette value *s*(*i*) is a measure of how similar an object is to its cluster (cohesion) compared to other clusters (separation). It is also possible to evaluate the discrimination through the overall average silhouette for the entire plot, which is the average of *s*(*i*) for all objects *i* in the dataset, known as the silhouette coefficient (SC) [26,27]. Briefly, the following interpretation has been reported for positive values of SC: strong (0.71–1.0); reasonable (0.51–0.70); weak (0.26–0.50); and no substantial classification has been found (≤0.25) [27].

## 3. Results and Discussion

### 3.1. Electrical Impedance Measurements in Distinct Flow Rates

Impedance measurements were performed in distinct flow rates, with a similar overall pattern. As a representative example, Figure 6 shows the impedance magnitude spectra of the collinear e-tongue for the sucralose-based sweeteners acquired at 5 mL/h. As all the sensing units are lined up in a single microchannel, three sets of impedance measurements were sequentially performed in all sensing units for each sweetener sample, resulting in efficient data acquisition. Figure 6 presents an average of three independent measurements with low standard deviation, indicating good repeatability of the sensor response. Despite the overall similarities, the sensing units show distinct impedance responses to each commercial brand of sucralose-based sweetener. However, it is challenging to evaluate the raw spectra, and in the following subsections, the post-processed data will be presented for better visualization and interpretation.

The device was manually handled only four times to get the results presented in Figure 6 to measure the four brands of sucralose-based sweeteners. We emphasize that using our previous microfluidic e-tongue setup comprised of one sensing unit per microchannel [9,11,21], 16 manual operations were required to analyze 4 sucralose-based sweeteners; while here, the number of manual operations corresponds to the number of analytes. Therefore, using the device presented here, the higher the number of analytes to measure, the more significant the reduction in operator manipulations and, consequently, the time of analysis. Moreover, the device was connected to the impedance analyzer only at the beginning of the data acquisition, with both multiplexing of the sensing units and data acquisition made automatically by software, instead of handling cables at the electrode’s pads between measurements. This specific manual step, besides wasting time, usually causes wear and tear of cables and IDE electrical contacts, which is now avoided. The exact reduction in the time of analysis is not trivial to estimate, as the manual steps are highly dependent on the operator’s skills.

As previously mentioned, our automated device enabled a systematic evaluation of the analytes in different flow rates. When we consider more than one flow rate analysis, a shift occurs in the impedance spectra, which can be more or less pronounced depending on both sensing unit and analyte. As a representative example, Figure 7 shows the impedance spectra for the PDDA/MMt-K sensing unit under different flow rates of Zero-Cal^®^ sucralose-based sweetener. Three independent measurements were acquired for each flow rate, and the average with low standard deviation is presented (the error bars are smaller than the symbol sizes). To evaluate if and how these shifts affect the analytes’ discrimination, we applied multidimensional projection techniques to the raw data, with the results discussed in the next subsection.

### 3.2. Principal Component Analysis for Electrical Impedance Data

Each analyte sample passed through twenty different flow rates. Therefore, we analyzed at least 35 PCAs, as some flow rates were evaluated more than once on different days to check the pattern’s reproducibility. Figure 8 shows representative examples of the processed impedance data in PCA plots for two distinct flow rates (1 and 5 mL/h). It was possible to distinguish among the four samples in both PCA plots, as well as for all the other PCA plots at the remaining flow rates that are not shown, meaning that the shift observed in the example of Figure 7 did not prevent the discrimination of these datasets. The successful distinction obtained with this robust e-tongue device for the sucralose-based sweeteners is probably related to the ingredients and manufacturing processes’ differences, as all samples rely on the same principal component (sucralose) and are nearly identical for the human gustatory system. It is not our intention to discover and discuss the root cause of discrimination among commercial samples, but to evaluate if it was possible to distinguish them, as they are sold as the same food product in the local market. Commercial sweeteners usually contain a plethora of additional ingredients, and these are sweetening substances, thickening agents, acidity regulators, flavoring, and preservatives, as clearly presented in the ingredients list of each product. This is because manufacturers need to focus not only on these excipients but also on other factors such as regional or national regulations and specifications of local markets.

Comparing Figure 8a,b, we can easily observe a lower dispersion and better visualization of the data in three independent measurements for the higher flow rate (5 mL/h). That is confirmed by the silhouette plots presented in Figure 9, with lower silhouette values for the 1 mL/h dataset, especially considering results for Finn^®^ sample (red bars in Figure 9a). Regarding the dispersion in data points on PCA (especially for Finn^®^ in Figure 8a), it is worth mentioning that the whole spectra of impedance magnitude versus frequency were used for multivariate data analysis, and the projection technique amplifies the dissimilarities between each spectrum, a fact that does not hamper the clustering of data. It is important to observe that the sum of the first two principal component yields 93.2% and 97.4% of the total information for measurements at 1 mL/h and 5 mL/h flow rates, respectively. However, since the heavier weight is on PC1 (above 80%), the plot analysis may be focused on PC1, so the sucralose-based sweeteners from Zero-Cal^®^ and Linea^®^ brands present more similar taste when compared to Finn^®^ and Adocyl^®^. This outcome is even more pronounced in the PCA plot obtained at 5 mL/h flow rate (Figure 8b). The quality of the discrimination was also evaluated through SC, obtained as an average of the values for the entire dataset presented in the silhouette plots (Figure 9). The calculated values of SC for PCA were 0.797 and 0.876 at 1 mL/h and 5 mL/h flow rates, respectively, meaning higher discrimination for the 5 mL/h dataset.

The results discussed so far are only selected examples of the entire experimental analysis. Figure 10 shows the plot of the SC values for the PCA plot at each flow rate, from 0 to 15 mL/h. All the SC values are above 0.78, meaning strong classification of the datasets using PCA, according to the standard suggested by Rousseeuw et al. [27]. Therefore, the shifts in the impedance spectra caused by changing the flow rate did not prevent discrimination in this case, but higher SC values facilitate recognizing distinct and similar groups by visual inspection. Moreover, there is an uptrend of SC by increasing the flow rate and stabilizing the values, reaching a plateau above ~3 mL/h. The possibility of fine-tuning the microfluidic e-tongue analysis by finding the best flow rate, besides achieving high discrimination, is interesting for optimizing the sample volume for measuring and discharge. This procedure is desirable for the evaluation of expensive analytes and biohazard samples. This conclusion is valid for this specific pair of microfluidic e-tongue and sucralose-based sweeteners. However, we believe it is possible to extrapolate this trend for analysis involving e-tongues with equivalent sensing units and analyte samples with similar viscosity. Therefore, more studies considering different devices and analyte samples (especially with higher viscosities) are suggested.

Figure 11 shows the PCA plot for the sucralose-based sweeteners considering 15 distinct flow rates (from 0 to 10 mL/h), and point out the clusters in the PCA plot is not straightforward like considering a single flow rate as we have done so far. According to previous results, Zero-Cal^®^ and Linea^®^ present more similar taste, when compared to Finn^®^ and Adocyl^®^. We have used *k*-means clustering to evaluate the PCA presented in Figure 11, and according to this method, three groups maximize SC, meaning Zero-Cal^®^ and Linea^®^ formed a single cluster indeed. The intersection occurs for Zero-Cal^®^ at higher flow rates and Linea^®^ at lower flow rates.

### 3.3. Analysis of Electrical Impedance Data by Equivalent Circuit Model

Impedance spectroscopy analysis allows the identification of potential interactions at the electrode/electrolyte interface through equivalent electrical circuits [28,29]. We used the model proposed by Taylor et al. of a device comprised of a metal electrode covered by a weakly conducting film in contact with an electrolyte interface [30]. The model was fitted to the experimental data using ZView software from Ametek Scientific Instruments (Berwyn, PA, USA) [31]. According to Taylor et al., the associated parallel resistance R_b_ and capacitance C_b_ are related to the film coating; the resistance R_t_ relates to the charge transfer across the film-electrolyte interface, i.e., through the double-layer; C_g_ is the associated geometric capacitance of the inter-electrode space filled with electrolyte. Finally, C_d_ represents the double-layer capacitance, which is charged from the solution resistance R_d_. The equivalent electrical circuit used for fitting the impedance data is shown in Figure 12.

Treating C_d_ as a perfect capacitor yields less than optimal fitting results. By converting it to a constant-phase element (Q_d_ from now on), we can yield a reasonable regression coefficient χ^2^ < 10^−3^. It can be modeled by an infinite network of associated resistors and capacitors transferring energy with a constant efficiency to the film-electrode assembly, synthesized by the constant-phase element contribution to the impedance [32]. Figure 13 shows the Bode plots (raw data and fitting) obtained for each sensing unit under a Zero-Cal^®^ aqueous solution at 5 mL/h flow rate to illustrate, as the fitting results for the sucralose-based sweeteners were very similar.

The fitting analysis was made mainly to evaluate the alterations in the double-layer capacitance values, Q_d_, at distinct flow rates. The exact comprehension of equivalent circuit analysis is far from simple, but it is a useful tool for better understanding such a complex system. For instance, no matter how close we get to 0 Hz, |Z| never reaches a plateau. It means that Q_d_ keeps growing and charging up via the solution resistance, represented by R_d_. It corroborates the fact that a thin ionic double-layer is formed over the electrodes [33,34]. Nevertheless, its growth rate is retarded, as shown in Figure 14 by a decrease in Q_d_ values by increasing the analyte flow rate. Therefore, the increase in the flow rate effectively shrinks the double-layer formation, revealing relevant physical parameters that effectively aid in the sensor response to the analyte. This effect is less pronounced for the bare IDE (sensing unit 1), indicating that the IDEs covered with LbL films are comparatively more sensitive to changes in the analyte flow rate than the bare one. Moreover, each sensing unit responded differently to the changes in the flow rate, meaning different double-layers are formed over each LbL film, as expected for sensing units covered with distinct materials.

The associated capacitances of the film coating (C_b_) and the geometry of the electrode (C_g_) dominate |Z| at higher frequencies (close to 1 MHz), masking the effects of other circuit elements. Moreover, the rapid upward trend of the phase angle as a function of the frequency compromises the sensitivity of the impedance measurement. Therefore, optimum sensitivity is achieved when the frequency is high enough to overcome the ionic mobility assisting the double-layer formation and low enough that C_b_ and C_g_ do not contribute expressively to the overall impedance. Therefore, the 100 Hz–10 kHz range is the most relevant for multivariate analysis in this sort of system, corroborating previously reported results [11].

### 3.4. Measurements of Sucralose-Based Sweeteners at Distinct Concentrations

Measurements of the sucralose-based sweeteners at different concentrations in water and in espresso coffee were carried out at a single flow rate of 5 mL/h, once it achieves both high discrimination and low discharge volumes.

#### 3.4.1. Analysis in Deionized Water

Figure 15 shows the PCA plot for the sucralose-based sweeteners dispersed in distinct concentrations in deionized water. The number in front of the sweetener brand represents these concentrations, from 2 to 10 μL/mL. Not all the dataset is labeled to avoid visual pollution, but only enough points are indicated to help visualize the overall trend. The white arrow indicates the increasing concentration, from right to left on PC1 for all the brands. On the other hand, the brands responded differently to PC2. The increased concentration of Finn^®^ is from top to bottom, while Zero-Cal^®^, Linea^®^, and Adocyl^®^ are from bottom to top. Another trend to highlight, also following the arrow direction, is the sequence of brands: Finn^®^, Zero-Cal^®^, Linea^®^, and Adocyl^®^. The highest concentration of Finn^®^ (10 μL/mL) intersect with the lowest concentration of Zero-Cal^®^ (2 μL/mL), and the lower concentrations of Adocyl^®^ (2 and 4 μL/mL) intersect with the higher concentrations of Zero-Cal^®^ and Linea^®^. While Finn^®^ and Adocyl^®^ clusters are more well defined, despite small intersections with other brands, the dataset of Zero-Cal^®^ and Linea^®^ merged. Considering them in the blue region, each respective concentration shows only small shifts from each other, as 8 μL/mL labeled in the plot as a representative example (Zero-Cal^®^ 8 and Linea^®^ 8).

Figure 16 shows the impedance measurements obtained for all the sensing units (bare IDE, PDDA/CuTsPc, PDDA/MMt-K, and PDDA/PEDOT:PSS) at 1 kHz from 2 to 10 μL/mL of sucralose-based sweetener brands. Three independent measurements were acquired at each concentration, and the average with low standard deviation is presented (the error bars are smaller than the symbol sizes for the most part). In these plots, the impedance data are represented by the square symbols and the lines are the linear fittings for each sensing unit. A decrease in the impedance magnitude as the concentration increases is observed for all the sensing units and sucralose-based sweeteners. Although the plots show the same trend, the absolute values of impedance are different for the sensing units according to the analyte, as expected from the operating e-tongue principle. Again, one of the goals here was the evaluation of the e-tongue ability to recognize such differences, especially considering the similarity among samples containing sucralose as the primary component.

From the linear fittings presented in Figure 16, it was possible to evaluate the limit of detection (LD), the limit of quantification (LQ), and the sensitivity of each sensing unit of the microfluidic e-tongue device for the sucralose-base sweeteners. The values of LD and LQ were calculated based on the IUPAC rule [35,36], LD = 3σ/S and LQ = 10σ/S, where σ corresponds to the standard deviation of the analysis, and S is the sensitivity obtained from the slope of the linear fitting. Table 1 shows LD, LQ, and sensitivity values obtained for all the sensing units to evaluate each sucralose-based sweetener. Briefly, the smallest LD and LQ for Finn^®^ was obtained with the PDDA/CuTsPc sensing unit, while the smallest LD and LQ for Zero Cal^®^ were with the PDDA/MMt-K. The PDDA/PEDOT:PSS configuration was the most reliable sensing unit for Linea^®^ and Adocyl^®^, presenting the most sensitive LD and LQ values. The different formulations of the sweeteners end up interacting distinctly with the films forming the sensing units, reinforcing the need for using materials with distinct physical properties to create them. Here, that was essential to both assist pattern recognition and improve the distinction of complex formulations apparently having the same taste.

Figure 17 shows the PCA plot for the neat and diluted sucralose-based sweeteners. The number in front of the sweetener brand represents these concentrations (from 2 to 10 μL/mL), while the labels without numbers represent neat commercial sweeteners (orange cluster). According to the *k*-means method, Zero-Cal^®^, Linea^®^, and Finn^®^ solubilized in deionized water were grouped in a single cluster (blue) according to the *k*-means method, while Finn^®^ is still the most different sample. Neat commercial sweeteners were grouped in the orange cluster, but Zero-Cal^®^, Linea^®^, and Adocyl^®^ data are closer to each other when compared to Finn^®^, corroborating the clustering for diluted sweeteners obtained by *k*-means. Moreover, the addition of neat sweeteners on the PCA confirms the trend already mentioned for the PCA plot presented in Figure 15: (i) increase in the concentration of sucralose; and (ii) the sequence of brands (Finn^®^, Zero-Cal^®^, Linea^®^, and Adocyl^®^), with both following the arrow direction.

#### 3.4.2. Analysis in Espresso Coffee

The sucralose-based sweeteners were evaluated in espresso coffee, considering 5 drops of sweetener in an espresso shot (10 μL/mL). The automated microfluidic e-tongue at 5 mL/h flow rate easily distinguishes all coffee samples tested with the sucralose-based sweeteners. Figure 18 shows the PCA plot for impedance data, with high SC (0.862) even considering a more complex matrix than water. Here, it is important to highlight that for the PCA analysis in aqueous solutions, Zero-Cal^®^ and Linea^®^ brands were closer to each other, therefore, delivering more similar taste; however, in the coffee matrix, the effect is not the same. This outcome may be related to the distinct abilities of each brand of commercial sucralose-based sweeteners to suppress the coffee bitterness.

All sweeteners were also analyzed in coffee from 1 to 5 drops (from 2 to 10 μL/mL). According to the manufacturer’s notes, five drops sweeten as much as one teaspoon of table sugar. Figure 19 shows the PCA plot for Zero-Cal^®^ at distinct concentrations in espresso coffee, as a representative example, with similar trends for the other brands. The white arrow indicates the increase in sucralose concentration, from right to left on PC1. The *k*-means method found four clusters, despite the five distinct concentrations evaluated. The algorithm grouped Zero-Cal^®^ at 6 and 8 μL/mL as a single cluster, once their distance is not as pronounced as from the other concentrations.

## 4. Conclusions

A new, robust e-tongue device comprising all sensing units in a single microchannel allowed the automated multiplexing of the sensor array. It reduced the number of manual operations four times compared to a previous system comprising one sensing unit per microchannel. Now, the operator only handles the e-tongue system between analytes. The exact reduction in the time of analysis is not trivial to be estimated, as the manual steps are highly dependent on the operator’s skills. However, for point-of-care application, for example, an analysis not too reliant on trained personnel is desirable, and we can assure that by reducing manual operations, the total time of analysis will be less dependent on that. Moreover, the device structure manufactured with standard industrial processes is easily adapted with emergent 3D-printing technologies for fast, robust, and portable instrument development for complex liquid analysis.

The device exhibited an excellent ability to discriminate among commercial sweeteners based on the same sweetening substance (four different sucralose brands available in the local market). The sweeteners’ successful differentiation is probably related to differences in the ingredients and manufacturing processes combined to deliver the final product. With the automated e-tongue, a systematic study of the influence of the analyte flow rate in the analysis was feasible and confirmed that increasing the sample flow reduces the double-layer formation, resulting in faster stabilization and better repeatability of the sensor response. The discrimination improved from static measurements to the analysis under flow conditions, reaching a plateau at ~3 mL/h. The possibility of fine-tuning the investigation by finding the best flow rate and achieving high discrimination is interesting for optimizing the sample volume for measuring and discharge. This procedure is desirable for the evaluation of expensive analytes and/or biohazard samples. This conclusion is valid for this specific pair of microfluidic e-tongue and sucralose-based sweeteners. However, it is possible to extrapolate this trend for analysis involving microfluidic e-tongues having equivalent sensing units and analyte samples with similar viscosity.

## Figures and Tables

**Figure 1 sensors-20-06194-f001:**
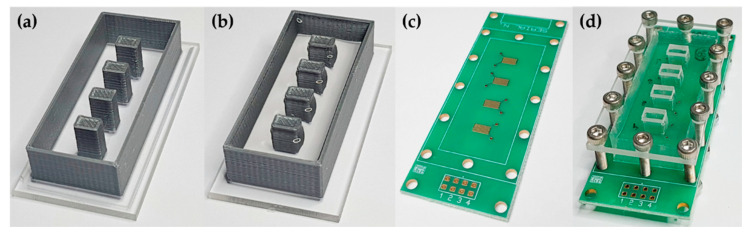
Development of the hollow structure to enable individual LbL deposition onto four collinear IDEs: (**a**) 3D-printed mold on an acrylic slide; (**b**) PDMS casting of the hollow structure; (**c**) PCB with the four collinear bare IDEs; (**d**) PDMS reversibly sealed on the PCB by mechanical pressure using screws and an acrylic plate.

**Figure 2 sensors-20-06194-f002:**
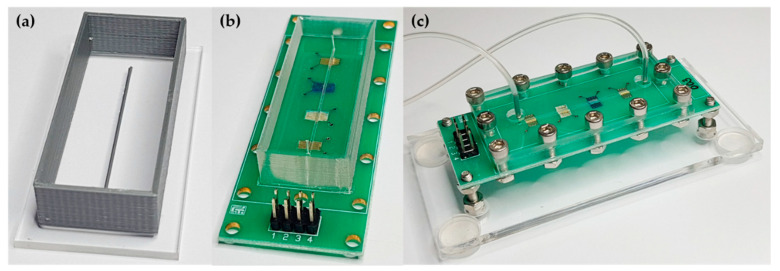
Steps of microchannel fabrication: (**a**) 3D-printed mold for PDMS casting; (**b**) PDMS microchannel aligned on the four collinear sensing units (IDEs covered with LbL films); (**c**) Final device comprised of four sensing units in a single microfluidic channel.

**Figure 3 sensors-20-06194-f003:**
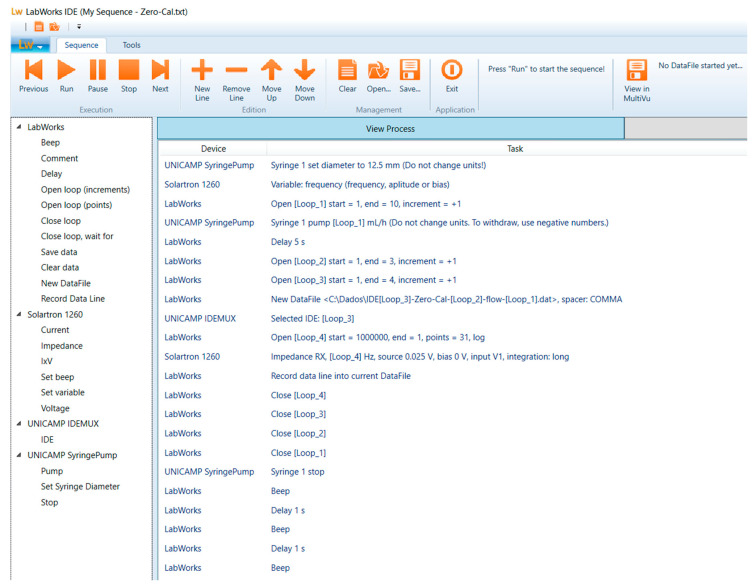
Software used for controlling the three instruments: custom-made syringe pump, Solartron impedance analyzer, and IDE’s multiplexer. An example of a sequence to automatically change the flow rate and multiplex the sensing units for electrical data acquisition is shown. Sequences can be easily adapted to meet distinct experimental needs.

**Figure 4 sensors-20-06194-f004:**
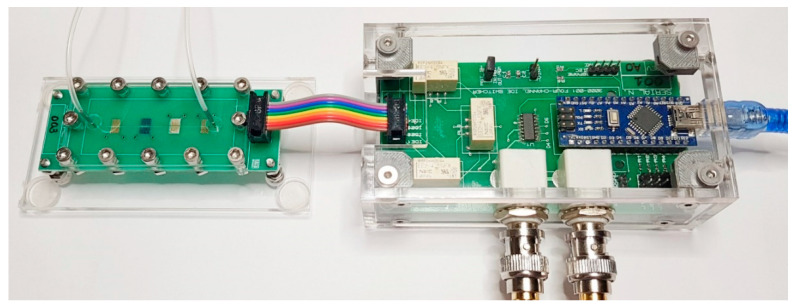
Final device for rapid impedance measurements: impedimetric e-tongue in a single microchannel (**left**) connected to the digitally controlled analog multiplexing system (**right**).

**Figure 5 sensors-20-06194-f005:**
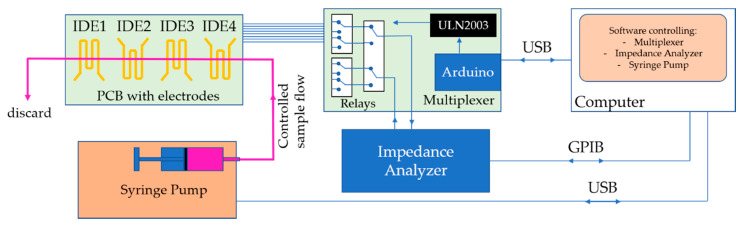
Diagram of the e-tongue setup comprising the microfluidic e-tongue device, syringe pump, multiplexer, and impedance analyzer, all automatically controlled by computer software.

**Figure 6 sensors-20-06194-f006:**
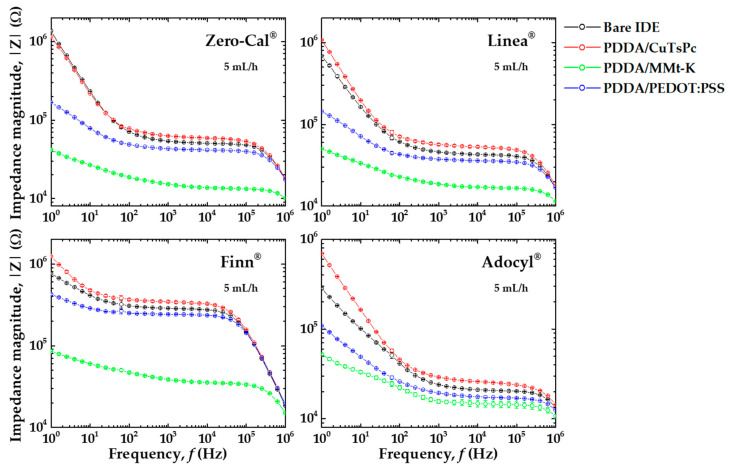
Impedance magnitude versus frequency spectra measured with the collinear sensing units (bare IDE, PDDA/CuTsPc, PDDA/MMt-K, and PDDA/PEDOT:PSS) in a single microchannel for the sucralose-based sweeteners of distinct brands (Zero-Cal^®^, Linea^®^, Finn^®^, and Adocyl^®^) in aqueous solutions (10 μL/mL) at 5 mL/h flow rate.

**Figure 7 sensors-20-06194-f007:**
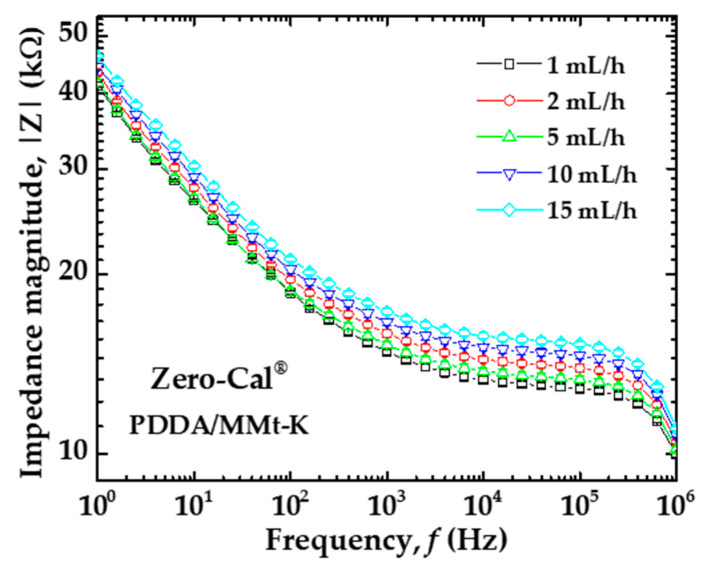
Impedance magnitude versus frequency spectra for the PDDA/MMt-K sensing unit under distinct flow rates of Zero-Cal^®^ sucralose-based sweetener in aqueous solution (10 μL/mL).

**Figure 8 sensors-20-06194-f008:**
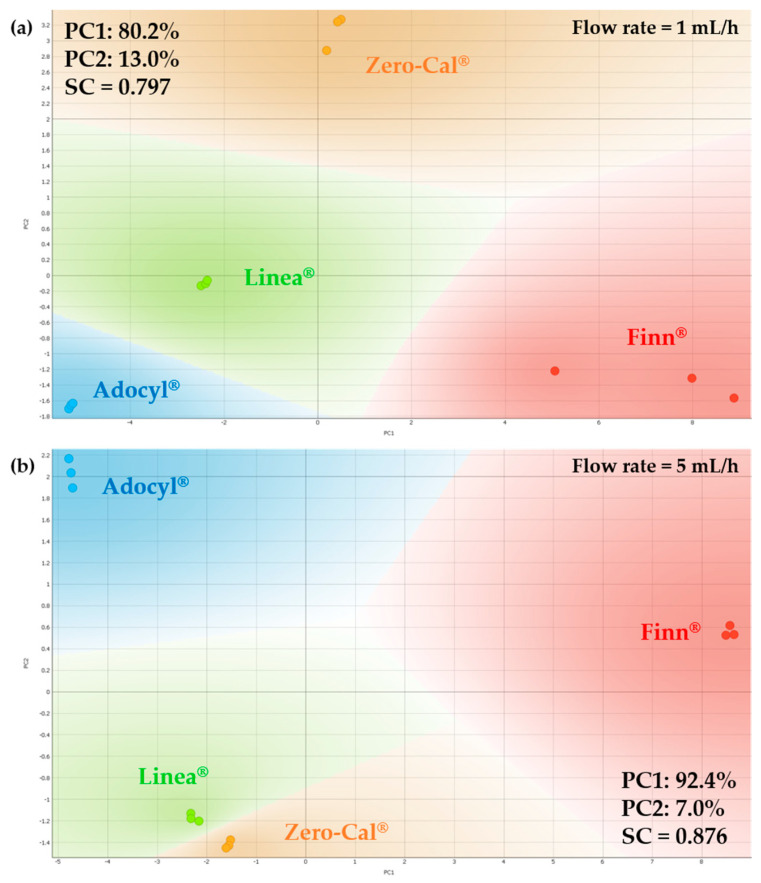
PCA plots for distinct brands of sucralose-based sweeteners in aqueous solutions (10 μL/mL) obtained using the impedimetric microfluidic e-tongue under (**a**) 1 mL/h and (**b**) 5 mL/h flow rates.

**Figure 9 sensors-20-06194-f009:**
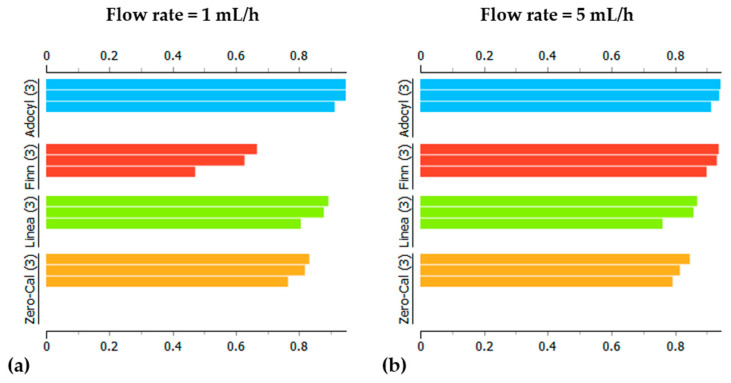
Silhouette plots of the sucralose-based sweeteners data obtained at (**a**) 1 mL/h and (**b**) 5 mL/h flow rates in a microfluidic e-tongue.

**Figure 10 sensors-20-06194-f010:**
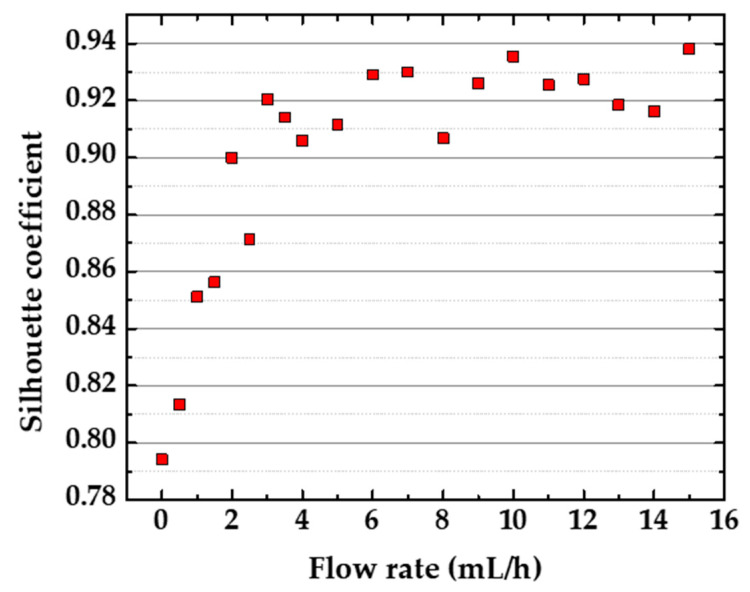
Silhouette coefficient (SC) for the PCA analysis of the impedance data for the sucralose-based sweeteners in aqueous solutions (10 μL/mL) obtained with a microfluidic e-tongue in distinct flow rates.

**Figure 11 sensors-20-06194-f011:**
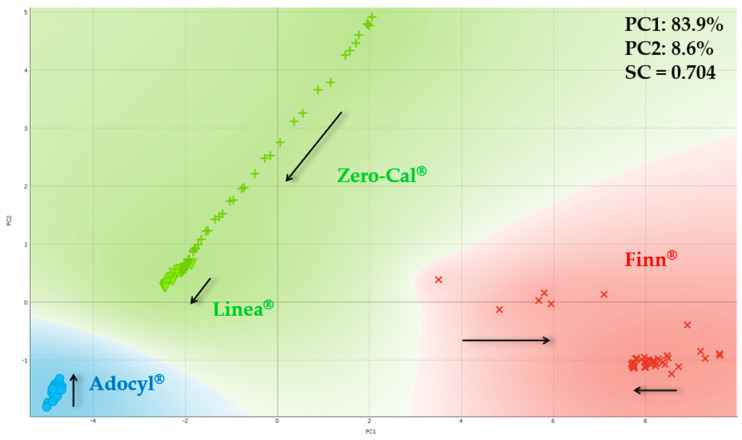
PCA plot for distinct brands of sucralose-based sweeteners in aqueous solutions (10 μL/mL) obtained using the impedimetric microfluidic e-tongue from 0 to 10 mL/h flow rates. The arrows indicate the direction of the increasing flow rate.

**Figure 12 sensors-20-06194-f012:**
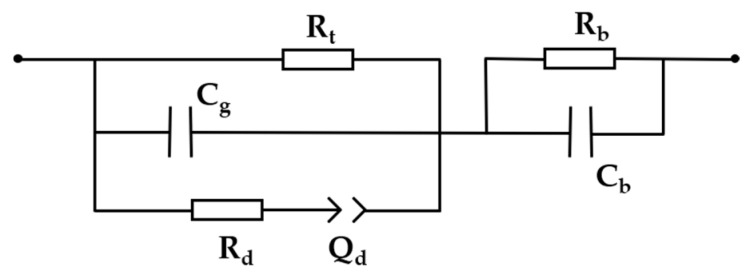
Equivalent electrical circuit used for fitting the impedance data.

**Figure 13 sensors-20-06194-f013:**
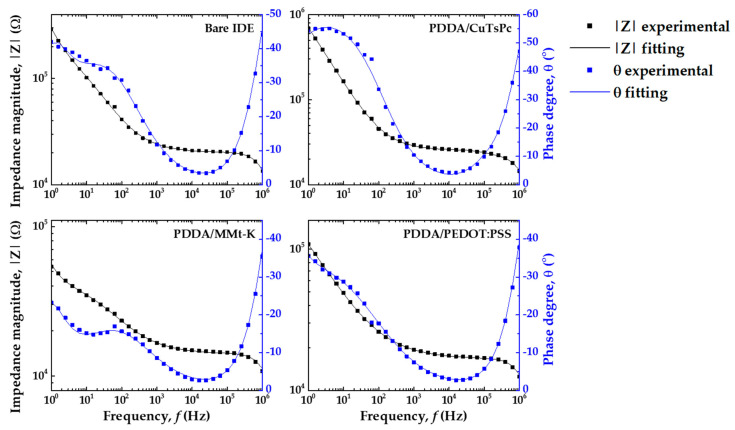
Measured (square) and fitted (line) Bode diagrams for bare IDE, PDDA/CuTsPc, PDDA/MMt-K, and PDDA/PEDOT:PSS sensing units in Zero-Cal^®^ aqueous solution (10 μL/mL) at 5 mL/h flow rate.

**Figure 14 sensors-20-06194-f014:**
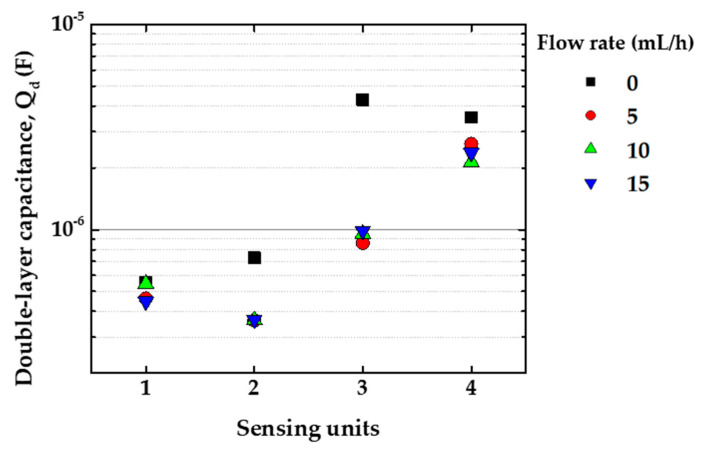
Double-layer capacitance, Q_d_, obtained by circuit equivalent fitting of each sensing unit for Zero-Cal^®^ in aqueous solution (10 μL/mL) from 0 to 15 mL/h flow rates. In the graph, the numbers 1, 2, 3, and 4 on the *x*-axis are related to the four sensing units: bare IDE, PDDA/CuTsPc, PDDA/MMt-K, and PDDA/PEDOT:PSS, respectively.

**Figure 15 sensors-20-06194-f015:**
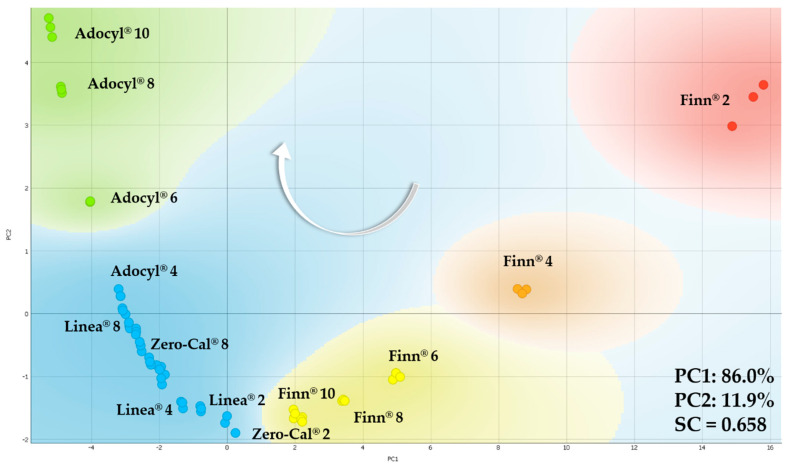
PCA plot for the sucralose-based sweeteners in distinct concentrations in deionized water (from 2 to 10 μL/mL) obtained using the impedimetric microfluidic e-tongue at 5 mL/h flow rate. The number in front of the sweetener brand represents their respective concentration. The white arrow indicates the direction of increasing concentration for all the brands tested.

**Figure 16 sensors-20-06194-f016:**
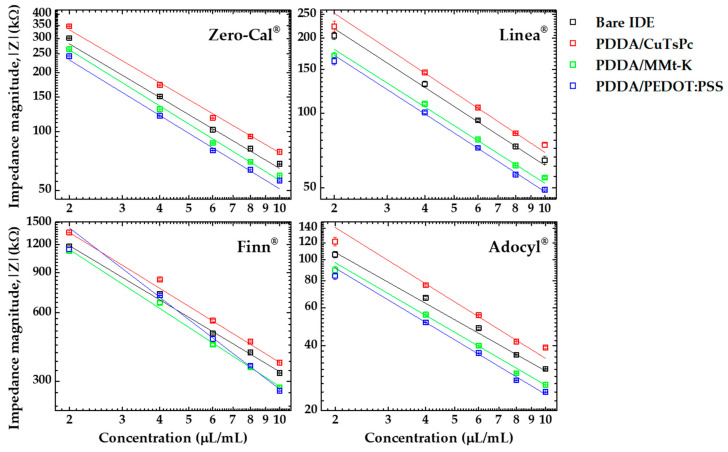
Impedance magnitude data at 1 kHz for each sensing unit evaluated at distinct concentrations (from 2 to 10 μL/mL) of four sucralose-based sweeteners using a microfluidic e-tongue under 5 mL/h flow rate.

**Figure 17 sensors-20-06194-f017:**
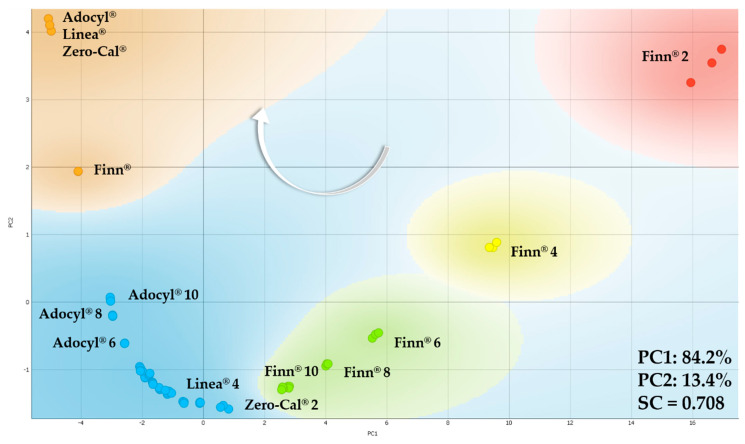
PCA plot for neat sucralose-based sweeteners and in aqueous solutions (from 2 to 10 μL/mL) obtained using the impedimetric microfluidic e-tongue at 5 mL/h flow rate. The number in front of the sweetener brand represents their respective solution concentration, while those without numbers represent the neat commercial sweetener. The white arrow indicates the direction of increasing concentration for all the brands tested.

**Figure 18 sensors-20-06194-f018:**
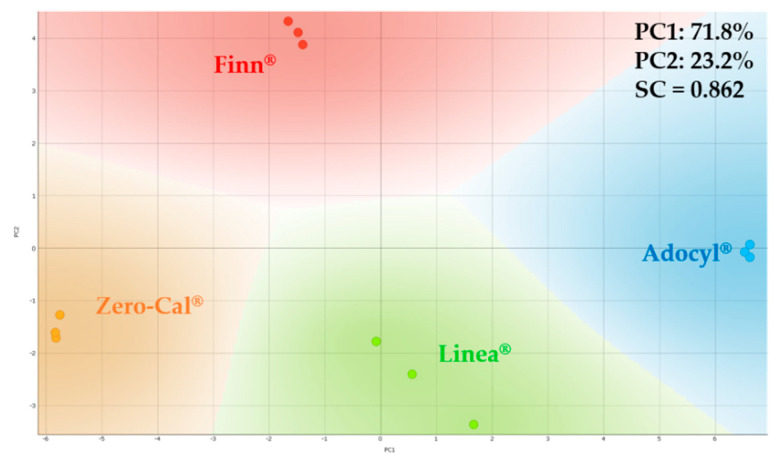
PCA plots for distinct brands of sucralose-based sweeteners in an espresso coffee matrix (10 μL/mL) obtained using the impedimetric microfluidic e-tongue under 5 mL/h flow rate.

**Figure 19 sensors-20-06194-f019:**
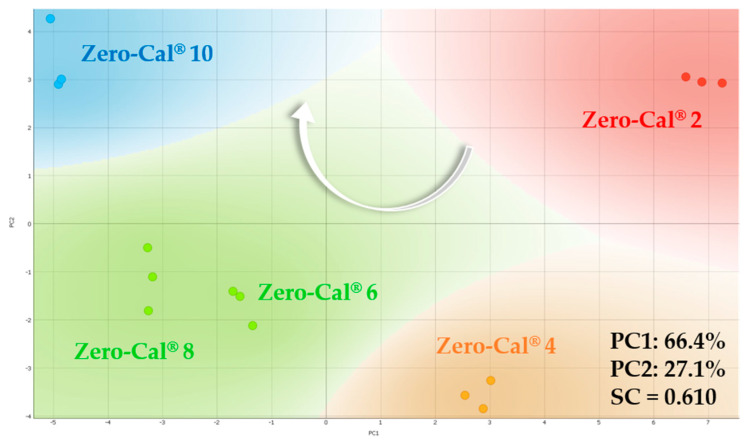
PCA plot for Zero-Cal^®^ at distinct concentrations in espresso coffee (from 2 to 10 μL/mL) obtained using the impedimetric microfluidic e-tongue at 5 mL/h flow rate. The number in front of the Zero-Cal^®^ brand represents the respective concentration and the white arrow indicates the direction of increasing concentration.

**Table 1 sensors-20-06194-t001:** Limit of detection (LD), limit of quantification (LQ), and sensitivity for the sucralose-based sweeteners using bare IDE, PDDA/CuTsPc, PDDA/MMt-K, and PDDA/PEDOT:PSS as sensing units in the microfluidic e-tongue device.

Sensing Units	LD, LQ, and Sensitivity *	Zero-Cal^®^	Linea^®^	Finn^®^	Adocyl^®^
Bare IDE	LD	0.12	0.11	0.05	0.21
	LQ	0.39	0.36	0.18	0.70
	Sensitivity	0.91	0.79	0.79	0.78
PDDA/CuTsPc	LD	0.10	0.11	0.02	0.25
	LQ	0.32	0.37	0.05	0.82
	Sensitivity	0.90	0.80	0.82	0.87
PDDA/MMt-K	LD	0.04	0.18	0.05	0.12
	LQ	0.12	0.60	0.18	0.41
	Sensitivity	0.96	0.77	0.86	0.81
PDDA/PEDOT:PSS	LD	0.10	0.09	0.11	0.11
	LQ	0.32	0.31	0.38	0.35
	Sensitivity	0.94	0.78	1.02	0.84

* The units for LD and LQ are (μL mL^−1^), and for sensitivity are (kΩ mL μL^−1^).

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
