# Peer review of "Influence of the Flow Rate in an Automated Microfluidic Electronic Tongue Tested for Sucralose Differentiation"

_sensors, 2020, doi:10.3390/s20216194_

Round 1
Reviewer 1 Report
After the corrections, the quality of resubmitted manuscript "Influence of the analyte flow rate in an automated microfluidic electronic tongue" has increased significantly.
I recommend accepting the article for further stages of evaluation after minor corrections:
- the tittle is too general. It must contain information about the application of the e-tongue is presented research.
- in Table 1 are presented calibraiton parameters LD, LQ and sensitivity. Please describe how these parameters were determined.
Reviewer 2 Report
The revised manuscript has been enhanced with a considerable amount of new data that allows assessing the performance of the developed sensing system. I have neither further comments nor remarks and suggest accepting the manuscript in its current form.
Reviewer 3 Report
The manuscript has significantly improved. There are however a few things that require attention.
- It was written in the methods section that "The e-tongue device was tested in four distinct brands of liquid sucralose-based sweeteners (Zero-154 Cal®, Linea®, Finn®, and Adocyl®)". There was no mention about testing different concentrations but this was presented in Figure 15 and Figure 17. The question now is, so what was the concentration of the samples in Figure 8, Figure 11.
- At the top left if Figure 17, adocyl, linea and zero-cal all have the sample orange color icon which, also happens to be the same for Finn. This should be clarified.
- Although authors introduced a new e-tongue system, there was no mention of drift. It is no secret that most sensor devices are susceptible to drift. In fact it is a primary challenge almost all sensor-based devices and apparent effects of drift could be seen in some of the results. E.g High inter-class distance (spreading of data points) for Finn in Figure 8A and Zero-Cal in Figure 11. Unless authors believe that their device is not susceptible to drift, it may be important to make mention of it as many mathematical drift corrections exist and could be beneficial for researchers in the future who may be interested in the new device introduced by the authors. Authors could check this paper for more details about drift and cite it if necessary: https://www.mdpi.com/2079-6374/10/7/74
Author Response
Please see the attachment.

This manuscript is a resubmission of an earlier submission. The following is a list of the peer review reports and author responses from that submission.
Round 1
Reviewer 1 Report
The submitted manuscript describes a modern and technologically advanced procedure employing impedance spectroscopy for differentiation between the different sucralose containing sweeteners.
Although the results shown in the manuscripts convince the reader that the developed sensing platform produced data that after a statistical treatment allows to discriminate commercial sweeteners, it is not particularly clear what is the major source of the differences observed. Beside the principal component, here sucralose, commercial sweeteners usually contain a plethora of additional ingredients and these are sweetening substances, thickeners, acidity regulators, flavouring and colouring matter as well as preservatives, to name a few. This is because manufacturers need to focus not only on these excipients which are appropriate when production capabilities and the technicalities of their manufacturing process are concerned, but other factors should be also allowed for, for instance, regional or national regulations, specifics of local markets, etc.
Therefore, a couple of questions arise, namely whether the developed procedure can distinguish between (i) different concentrations of sucralose in the same matrix, and (ii) different sweeteners in the same matrix.
Finally, in the conclusion section the Authors stated: “The device exhibited excellent distinction ability to separate brands of sucralose-based sweeteners supplied from different producers, instead of different sweeteners and masking effects commonly found in the literature.” It is suggested that clarification to this statement is added as it is not evident how this sentence should be understood. Likewise, more light should shed on “masking effects” mentioned in the text, for this might be perceived by some readers as another example of ambiguity.
Reviewer 2 Report
The manucript presents the automated impedimetric electronic tongue used in microfluidic channel dedicated for sucralose-based commercial sweeteners detection.
The experiments, e-tongue setup and methods are presented clearly. However, the reviewer asks the authors to respond to the following issues:
- The tittle suggest that constructed electronic tongue was able to detect the sucralose-based sweeteners. In fact the presented study shows only the possibility of discrimination between four commercial sweeteners. It seems you should replace the word "detection" for "discrimination" in the tittle.
- Line 142: "Aqueous solutions of each brand were prepared at 10 mg/mL, the equivalent of ca. 5 drops in one shot of espresso coffee". Presented results refer to aqueous solutions. It is the simplest matrix. Is it even useful to distinguish between these sweeteners in pure water? If the matrix was espresso coffee, would the electronic tongue still be useful? In my opinion, the results of research on real samples (not only model solutions) must also be presented.
- Line 195: "... PCA plot of real capacitance data collected at 1kHz..." The capacitance at 1kHz is one dimensional value. Please clearly indicate which values were used for PCA plots preparation.
- In Figure 6 it is shown that PC1 explains over 98% of the data set variability. With such high values, a more in-depth PCA factor analysis is needed. Please check if any of the variables has a big factor loading on PC1.
- Please remove "Supplementary materials" section from manuscript
Reviewer 3 Report
This aimed at developing a microfluidic-based electronic tongue system using LBL films for identifying commercial sweeteners containing sucralose. Short-time measurement, low-cost measurement system, and reduction of sample volume are necessary for e-tongue systems. Unfortunately, I regret to inform you that your manuscript can not be considered for publication in the “Letter” of Sensors because the paper does not propose new sensor concepts from the authors’ previous papers. This paper describes a plurality of contents such as the method of forming the LbL, the flow channel design, and the analysis method, but none of them are sufficient explanations. I recommend narrowing down to one and posting in a full paper. In that case, the authors should clear and consider the advantages of using the microfluidic device (maybe the result does not change even in batch measurement?), basic characteristics to sucralose of LbL (should include pure sucralose to PCA analysis) or sensor performance compared with the conventional method, etc. for sucralose analysis performed sufficiently.
Reviewer 4 Report
The study aims to develop an impedimetric e-tongue through affordable means. This is a particularly important subject for food quality control and quality assurance. The figures in the study also suggests that the device is very portable, this is an extra advantage, however, certain parts of the manuscript can be improved. Detailed corrections are in the uploaded manuscript but below is a summary of some things that require attention.
The whole manuscript should be thoroughly checked for English discrepancies.
Introduction: The study involves much technicality about development of an e-tongue so I was naturally expecting to see some brief details about this in the introduction: what are the general requirements? what challenges can be expected? Etc. In addition, lines 66-70 seems like methods/findings and does not fit in the introduction.
Materials and methods: Authors used certain compounds for the development of their e-tongue: copper phthalocyanine-3,4’,4’’,4’’’- tetrasulfonic acid tetrasodium salt (CuTsPc), montmorillonite clay (MMt-K), poly (3,4- ethylenedioxythiophene)-poly(styrenesulfonate) (PEDOT:PSS), and poly(diallyldimethylammonium chloride) solution (PDDA). Why were these particular ones chosen?
A major disadvantage of the e-tongue is drift which, can be caused my many conditions during the experiment. How were the samples analyzed? It will be nice to have a sequential description of your analytical procedure. This could be in the form of a figure and, would also help understand the rapidness of your instrument and analytical procedure.
Was there any rinsing in between sample measurements? How was cross-contamination avoided?
Results and discussion: It were written in line 168 that “Only one set is presented below to avoid redundancy of information, as all three independent measurements displayed similar behavior”. How about presenting an average of the three independent measurements, this is statistically sound and a good way to check the repeatability. In figure 5, There is something happening at the frequency of 10-3, in all the plots. Could you explain that? The lines are distinguished by both color and shape in the plot, this should be reflected in the legend at the top right of the plots as well.
